# Tracing the evolution of green logistics: A latent dirichlet allocation based topic modeling technology and roadmapping

**Ruijundi Ma, Yong Jin Kim**⊙*

Graduate School of Logistics, Inha University, Incheon, South Korea

* yongjin@inha.ac.kr

**Data Availability Statement:** The DOI of the code and data is DOI: 10.5281/zenodo.8116024 (https://zenodo.org/record/8116024).

**Funding:** This work is supported by the Korea Agency for Infrastructure Technology

## Abstract

Green logistics (GL) is the main development trend of modern logistics. The analysis of green logistics topics and their evolution is helpful in grasping its development trend and doing research facing the international frontier. Focusing on the hot topics and evolution process of green logistics, this paper analyzes from four aspects: firstly, this study divides the green logistics development progress into three stages based on life cycle theory, which are the emerging stage (1993–2003), slow growth stage (2004–2014) and rapid growth stage (2015–2021). Then, based on latent dirichlet allocation (LDA) topic model, this study summarizes and confirms related words and meaning of each topic in different stages. Furthermore, this study calculates the text similarity in each development stage of green logistics and analyzes the trend of hot topics in green logistics. Finally, this paper visualizes the development roadmap of green logistics and explores the progression among three stages. There are 4, 5, and 7 topics defined respectively in three development stages. The revolution of green logistics is analyzed, and the results show that "model and management on sustainable development of GL", "related issues and potential threats of GL", and "optimization analysis of low-carbon vehicle routing and time" are the primary development trends of green logistics. This study fills the gap in considering the evolution process of green logistics through topic modeling and roadmapping method. It provides a particular theoretical significance for the green and sustainable development of logistics.

## 1. Introduction

Green logistics plays a vital role in the metabolism of modern logistics, which is profoundly impacted by climate change and sustainable development. As data of the International Energy Agency (IEA) shows that the $CO_2$ emissions of the transportation sector is 7098 million tons in 2020, which is 2727 million tons more than in 2000. Saving resources and protecting the environment are the basic concepts of green logistics. The main difference between green logistics and traditional logistics is that scrap can be recycled and reused to a certain extent. Moreover, as transportation is essential to logistics, adopting electric vehicles and alternative fuels is also the key to developing green logistics. With the popularity of sustainability, topics related to the development of green logistics have been integrated into diverse logistics functional departments.

Advancement (KAIA) grant funded by the Ministry of Land, Infrastructure and Transport (Grant 21HCLP-C161726-01). The funders had no role in study design, data collection and analysis, decision to publish, or preparation of the manuscript.

**Competing interests:** The authors have declared that no competing interests exist.

Extensive research has shown that different issues related to green logistics require long-term attention and study [1]. Improving cost efficiency, speeding delivery, and effectively reducing carbon emissions have become common goals in the logistics industry [2]. However, the subdivided logistics sector has led people to get used to looking at the development of green logistics from a single perspective. For example, formulating daily environmental management principles [3], strengthening effective collaboration among multiple departments [4], optimizing transportation routes [5], developing self-driving vehicles [6], purchasing low-consumption vehicles [7], and environmentally friendly packaging materials [8] are often discussed. Therefore, it is necessary to trace the evolution of green logistics and obtain the route of its central topics' development from a macro perspective so as to clarify the direction its future tendency will take.

Extracting the topics in the evolution process of green logistics is the clue to analyzing macro issues. Topic mining has been an influential evolutionary analysis method in recent years, which can be used in disparate industries. Kleinberg (2003) [9] proposed that hot topics can be mined by using the characteristics of word frequency distribution. However, traditional topic analysis methods, such as word co-occurrence analysis, are not enough in the face of the rapidly increasing amount of textual information with rich content nowadays. Among the mainstream topic models in topic mining, the Latent Dirichlet Allocation (LDA) model is widely used to identify latent topics in the initial document. Since journals are important scientific research achievements, it is trustworthy when the journal content is analyzed by topic mining. Silva and Moro (2021) [10] present a text-mining literature analysis of published articles from Scopus database by LDA topic model regarding blockchain technology and consumer trust. Paul Tae-Woo Lee et al. (2020) [11] conduct a literature review on the logistics, supply chains, and transport (LST) field through 388 English and Chinese papers by LDA and bibliometric analysis. The topic model based on text analysis has become increasingly crucial for evolutionary analysis.

This study performs a topic mining and evolution analysis of green logistics based on literature data, describing the progressive path of sustainable development in logistics. We have devised a methodological approach to elucidate the relationship among the major topics in diverse phases of green logistics. This mixed methodology incorporates prior research insights, leveraging the life cycle theory, the latent dirichlet allocation (LDA) topic model, and text similarity calculations. According to the article data, this study divides green logistics into different time-window stages by life cycle theory. It utilizes the LDA topic model to interpret the key topics under specific time windows. Then, the similarity of various topics in each stage is measured by cosine distance value. Finally, a visualization method is used to show the evolution of green logistics. This method can efficiently state the research and development (R&D) status of green logistics. In short, this study objectively reveals the research topic of green logistics and its evolution process and looks forward to the development trend of green logistics in the future.

## 2. Literature review

There are three dimensions of the existing literature related to our research work, which includes the status quo of green logistics, topic modeling technology and roadmapping analysis.

### 2.1 Status quo of green logistics

Due to the reality of global warming, modern logistic companies are required to reduce greenhouse gas emissions such as carbon dioxide ($CO_2$) to comply with environmental regulations.

It is inevitable for the development of green logistics to protect the environment from further damage. There is no unified definition for green logistics in the industry. Lee and Klassen (2008) [12] think that green logistics is similar to green supply chain management and define it as supply chain management considering environmental factors to improve environmental performance. However, green logistics and green supply chain should be considered separately. Sbihi and Eglese (2010) [13] point out that the concept of green logistics refers to the production and distribution of goods, taking into consideration environmental and social factors in the process. In addition, green logistics is also considered a closed loop incorporating traditional forward logistics with reverse logistics, which creates a recoverable product environment [14]. Generally, forward logistics is the transition of goods from raw materials to end consumers, including material purchasing, warehousing, manufacturing, transportation to distributors, and final delivery to consumers [15]. Therefore, we classify green logistics by combining forward and reverse logistics to gain a deeper understanding of its scope. The framework of green logistics is shown in Fig 1, which includes green purchasing [16], green warehousing [17], green production [18], green transportation [19], green delivery [20], and reverse logistics [21].

Moreover, previous research has established that additional subgroups can be subdivided from the six parts of green logistics. The existing literature on green purchasing focuses particularly on green purchasing policy (GPP) [22], green purchase intentions [16], and factors influencing green purchase intention [23]. Green warehouse management [24], environmental impact of warehouse building [25], and energy-saving in warehousing [17] are the main parts being analyzed in green warehousing. A great deal of previous research on green production has focused on intelligent manufacturing [18], real-time production logistics [26], and green packaging [6]. Green transportation can be classified into an assessment of transportation system [27], green transportation strategy and challenge [28], and key determinants of sustainable transportation [29]. The critical aspects of green delivery can be listed as follows: optimal delivery strategy [30], last-mile delivery [31], green vehicle routing problem [32], and assignment of green delivery [33]. Reverse logistics is mainly studied from green recycling strategy [34], structure [35], critical factors [36], and optimal model [37]. These studies indicate that green logistics is at the peak of its development, affecting every aspect of the industry.

It should be noted that low carbon emissions have become an essential indicator of the environmental efficiency of logistics processes. Consequently, low-carbon technologies are

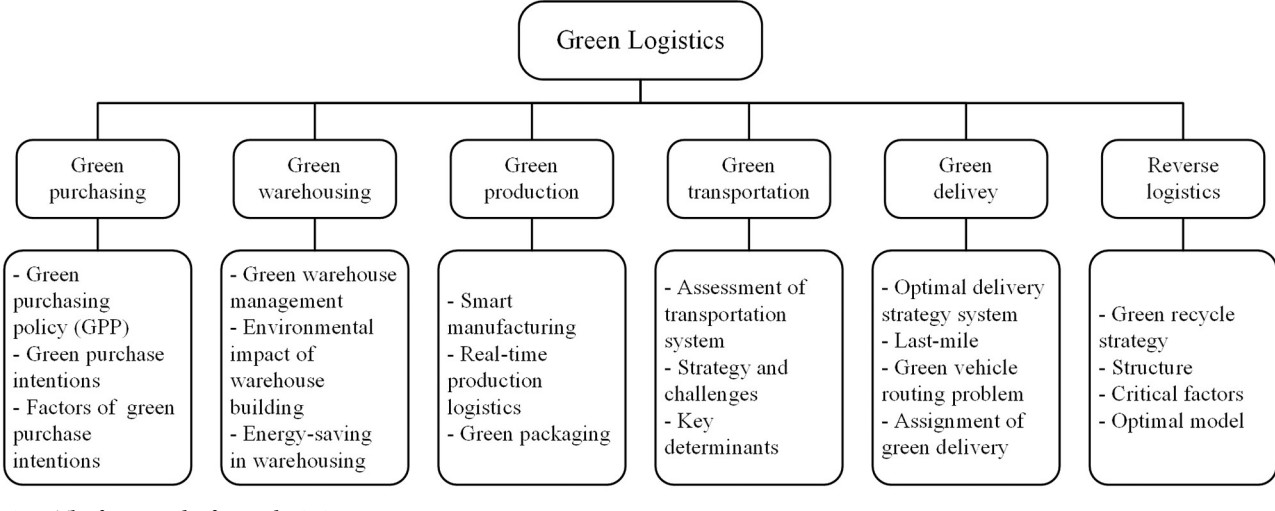

**Fig 1. The framework of green logistics.**

also becoming increasingly popular. To reach carbon neutrality, Bai S et al. (2022) [38] examined the R&D investment efficiency of the lithium battery industry in China. A survey from Shang and Lv (2023) [39] shows that the achievement of carbon neutrality requires considerable effort, including the development of improved low-carbon technologies. Additionally, the government's carbon emission policy plays a crucial role in fostering the growth and progress of green logistics [40, 41]. While a considerable body of research has been carried out on the details of green logistics process, much less is known about the macro perspective of green logistics development. It fully shows that analyzing the evolution of green logistics is substantial.

## 2.2 Topic modeling technology

Topic modeling is a popular analytical tool for evaluating data, initially developed in the 1980s. Various sources and forms of data can be analyzed by topic modeling, like images [42], spatial data [43] and bioinformatic data [44]. For the sake of simplicity, text-based data will be discussed in this paper. Topic modeling is an unsupervised machine learning technique, and we can determine cluster words for a set of documents by automatically analyzing text-based data [45]. Many scholars use topic modeling to deal with complex datasets because of their flexibility and adaptability. There are four most popular techniques today, which are Latent Semantic Analysis (LSA), Probabilistic Latent Semantic Analysis (PLSA), Latent Dirichlet Allocation (LDA), and the newer, deep learning-based lda2vec. LDA is the most popular and generally effective topic modeling technique compared to other methods [46].

Much of the literature on LDA pays particular attention to determining vital topics for different research fields. Zhang H et al. (2021) [47]used the LDA topic model to perform technology assessment and roadmapping analysis of the blockchain field based on patent data. Du et al. (2020) [48] improved LDA into the topic extraction model (named MF-LDA) to accurately extract topics from Microblog posts. Bastani K et al. (2019) [49] analyzed the Consumer Financial Protection Bureau (CFPB) consumer complaints based on LDA. Moreover, the LDA topic model can be used to study road safety problems [50] and healthy problems [51].

It has previously been observed that bibliometrics is widely used in analyzing literature [52]. Bibliometrics is a statistical method for quantitative analysis of specific research fields proposed by Alan Pritchard in 1969. It includes citation analysis, co-citation analysis, bibliographic coupling, co-author analysis and co-word analysis [53]. Many researchers have utilized the LDA topic model to analyze the text for a set of documents. The corpus's topic features can be obtained through machine learning and perform mining, classification, and refinement. Therefore, we introduce the LDA topic model into the article analysis of green logistics.

## 2.3 Roadmapping analysis

In the 1970s, Motorola developed the roadmapping approach as a way to align technology and innovation, which is widely adopted by companies, governments, and other institutions [54]. The flexibility and scalability of roadmapping allow for different roadmaps to be customized for the research object and innovation environment, which facilitates visualizing the process [55]. For example, Lu and Weng (2018) [56] integrated a maturity model and technology roadmap to describe emerging technology development trends. Zhang H et al. (2021) [47] combined LDA and roadmapping based on patent data to analyze the field of blockchain. Nazarko et al. (2022) [57] consolidated roadmapping, technology mapping and scenarios to describe the application of nanotechnology foresight.

It is common to use roadmapping in technology fields, but it has been extended to a much broader range of areas. According to Ma T et al. (2006) [58], roadmapping can serve as a

practical approach to knowledge management in academia and as a means to facilitate scientific research. Rivkin et al. (2019) suggest a roadmap for future research aimed at advancing the ecological understanding of urbanization [59]. The role of roadmapping in green logistics is less considered. Hence, we aim to introduce the roadmapping method to visualize the evolution of green logistics, encompassing the development of environmentally friendly products, green supply chain technology, policies, company performance, and low-carbon transportation.

All of the studies reviewed here show that many scholars have done much research in green logistics, the LDA topic model and roadmapping severally. Few studies use article data and topic model to systematically analyze the evolution of green logistics fields and forecast its future development trends. To address this gap, we use article data to analyze the green logistics field and build an automated analysis method based on LDA topic model to describe the different stages of green logistics development. Finally, we can visualize the evolution process of green logistics by roadmapping and forecast its future development trends.

## 3. Methodology

Web of Science (WoS), the world's oldest, most widely used and authoritative database of research publications and citations, has expanded its selective, balanced, and complete coverage of the world's leading research to over 34,000 journals [60]. Therefore, the initial article data was obtained from the Web of Science and searched with no restrictions relating to academic disciplines and journals to pledge to describe the whole development process of green logistics credibly. The division of green logistics development phase depends on the life cycle theory. After division, the specific development stages of green logistics are analyzed respectively. The article data needs to be preprocessed before LDA topic model, including text segmentation, removing stopwords, and stemming. Then, we can get "document and topic matrix" and "topic and word matrix" by LDA topic model, and topics can be determined for each evolution stage. For the green logistics evolution of analysis deeply, the topic and word matrix in different phases are calculated by text similarity, which is indicated by Cosine formula. The visualizations of topics in each green logistics evolution stage are realized by LDA-vis. Finally, we can get the roadmapping of green logistics evolution based on the results of text similarity. The LDA-based research method flow can be seen in Fig 2.

### 3.1 Division of evolutionary stages

This study uses the life cycle and S-curve as a basis for the division of green logistics evolution. Life cycle theory was first proposed by Karman in 1966, and it was developed by Hersy and Blanchard in 1976. The standard life cycle thinks that the industry generally has four phases: the emerging period, growth period, maturity period and decline period [61]. Since the actual industry is more nuanced, it allows us to gain a deeper understanding of it and hedge against its future crises. The patterns of emerging, growth and decay can be illustrated visually by S-curve. Many systems have natural life cycle like organization development, technology development, phases of innovation and social phenomena. Nothing that grows can continue to grow indefinitely. The growth dynamics of a typical product lifecycle follow a "bell-shaped" curve. Despite life cycle theory's widespread application in describing the development and evolution of products, industries, and organizations, article data is not directly related. Nevertheless, we can analyze and explain the development and evolution trend of the industry represented by article data using the thinking frameworks and concepts of life cycle theory. Therefore, the scientific and potential patterns of green logistics evolution can be revealed using the life cycle theory and S-curve.

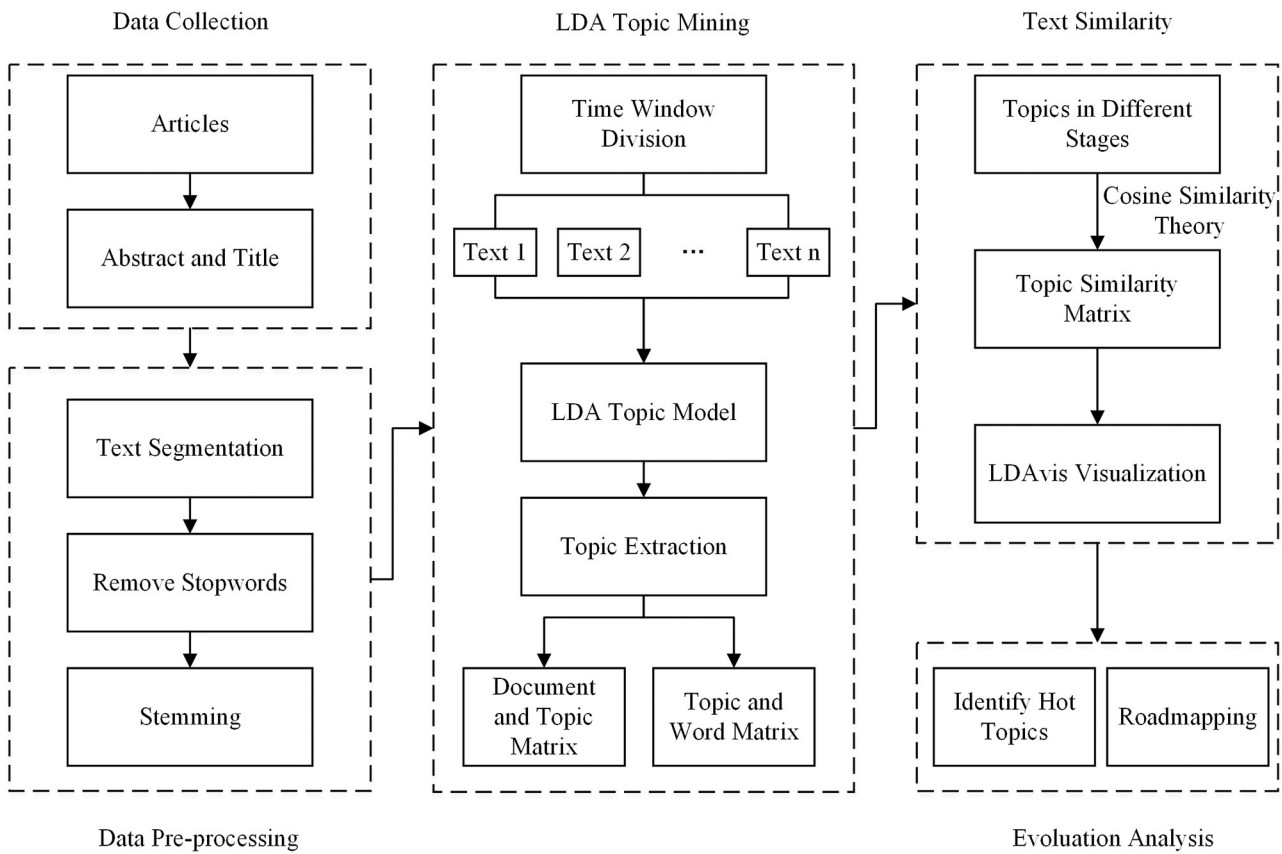

**Fig 2. LDA-based research method flow.**

### 3.2 LDA topic model

Topic modeling is a critical technique in text mining, extensively employed in various domains such as data mining, text classification, and community discovery. The topic model has gained significant traction in the field of natural language processing due to its exceptional dimensionality reduction capabilities and its flexible and extensible nature in constructing probabilistic models. The LDA (Latent Dirichlet Allocation), proposed by Blei et al., is a notable probabilistic topic model where topics are treated as probabilistic distributions of terms [62]. The LDA topic model extracts topics from text, a document topic generation model that contains a three-layer Bayesian network of words, topics, and documents. LDA uses prior distribution to overcome the problem of overfitting in the learning process.

Moreover, two aspects can be considered in order to evaluate the advantages of the LDA method in green logistics evolution analysis. On the one hand, one of the most well-known methods for assessing article data is bibliometrics, which visually analyzes the citation, co-citation, bibliographic coupling, co-author, and co-word through software such as CiteSpace and VOSviewer [53]. However, based on R programming and Python algorithms, the LDA method allows for analyzing hot topics for each stage in green logistics development, which is essential for this research. On the other hand, several topic modeling techniques are available, but LDA is the most prominent and typically influential among them. Specifically, LDA is generally more effective than PLSA (Probabilistic Latent Semantic Analysis) because it can generalize easily to new documents. In addition to being a probability generation model, LDA is easily extensible to other models, demonstrating its flexibility [46].

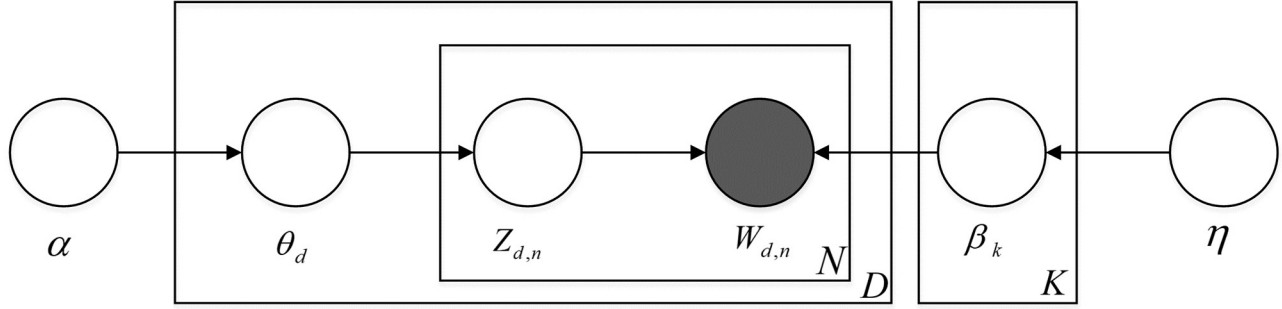

**Fig 3. Structure for LDA topic model.**

There are three assumptions in LDA topic model, which include that: topics are represented by multinomial distributions of words; documents are characterized by a multinomial distribution of topics; topic-word distribution and document-topic distribution, the prior distributions of both are Dirichlet distributions [45]. Since the Dirichlet distribution is a conjugate prior distribution of multinomial distributions, the document-topic distribution and topic-word distribution can be speculated through the observed word sequence, and the latent topics can be mined afterward. A structure for LDA topic model [62] is shown in Fig 3. Nodes represent random variables. Solid nodes represent observed variables, and hollow nodes represent hidden variables. Arrows indicate probabilistic dependencies. The rectangles represent repetitions, and the numbers inside the rectangles represent the number of repetitions. Table 1 shows the explanations of parameters in LDA topic model.

The process of LDA topic model can be described briefly as follows. Firstly, a set of documents are selected from the database. Then, we sample from a prior distribution with parameter $\alpha$ and a prior distribution with parameter $\eta$ to generate the distribution of documents over topics ($\theta_d$) and the distribution of topics over words ($\beta_k$). Finally, we sample $Z_{d,n}$ and $W_{d,n}$ from multinomial distributions with $\theta_d$ and $\beta_k$, respectively. The parameter estimation process of the LDA topic model is to estimate the value of the hidden variable based on the value of the observed variable. This study uses the Gibbs sampling algorithm to estimate the parameters $Z_{d,n}$, $W_{d,n}$ of the LDA topic model.

### 3.3 Text similarity and visualization

Text similarity calculation is the process of assessing the similarity between two or more entities using a method that produces a quantifiable similarity value [63]. The text similarity

**Table 1. Explanations of parameters in LDA topic model.**

| Parameter | Explanation |
|---|---|
| D | All documents in the corpus |
| N | The number of words in each document |
| K | Total number of topics |
| $\alpha$ | Proportion parameter |
| $\eta$ | Topic parameter |
| $\theta_d$ | Distribution of topics on document $d$ |
| $\beta_k$ | Distribution of words on topic $k$ |
| $Z_{d,n}$ | The subject of the $n_{th}$ word in document $d$ |
| $W_{d,n}$ | The $n_{th}$ word in document $d$ |

algorithm is essential for text mining. This study uses cosine similarity to give similarity between two topics in terms of their subject matter. The similarity of the results ranges from -1 for complete opposites to 1 for absolute sameness, 0 for orthogonality or decorrelation, and intermediate values for intermediate similarity or dissimilarity.

The residual string similarity, denoted as θ, between two attribute vectors A and B can be calculated using the dot product of the vectors and their respective lengths, as expressed below:

$$\text{similarity} = \cos(\theta) = \frac{A \cdot B}{\|A\|\|B\|} = \frac{\sum_{i=1}^{n} A_i \times B_i}{\sqrt{\sum_{i=1}^{n} (A_i)^2} \times \sqrt{\sum_{i=1}^{n} (B_i)^2}} \tag{1}$$

When we calculate the similarity between two texts, the attribute vectors A and B are word frequency vectors in the documents. Based on the word frequency matrix of each topic, we can calculate the similarity between two topics through cosine, also called cosine similarity.

This study can visualize the evolution stages of green logistics based on the results of the LDA topic model, which is realized by R programming using the LDAvis package. As can be seen in the visualization, there is a clear distinction between each topic in terms of strength and degree of discrimination. Moreover, the evolutionary relationship between different topics at various stages can be represented by arrows. The evolution and development of key topics are critical for an industry, which is conducive to analyzing the future trends of the industry.

## 4. Experiment and results

### 4.1 Data collection and data pre-processing

This study collects article data from Web of Science (WoS) and selects advanced search function. The word "green logistics" was used as the search term, the "topic" was used as the search route, and the search year is from 1993 (the earliest year on this website) to 2021. Then, we can export literature records that include the "abstract" and "title" to plain text files and filter out duplicate and incomplete documents. Finally, 3437 articles were collected.

After obtaining the required data, the cumulative number of articles each year is calculated. The result can be seen in Fig 4. The growth tendency of green logistics articles indicates that the number of articles published in the early years is low. As green logistics gains traction and popularity, the number of papers will continue to increase from a low growth rate to a rapid growth rate as the trend continues. Thus, to analyze the topic and evolution of green logistics better, we can divide the development of the green logistics field into three stages based on the life cycle theory and S-curve, which are the emerging stage of 1993–2003, the slow growth stage of 2004–2014, and the rapid growth stage of 2015–2021.

From the three stages of green logistics article data, we can generate three datasets for topic mining and visualize the results. There are 141 articles in the emerging stage, 810 articles in the slow growth stage, and 2486 articles in the rapid growth stage. Before the LDA topic model, we need to preprocess the three datasets. We filter the literature plain text file through format conversion to get three documents, including only the "title" and "abstract". These three documents are fundamental datasets before R programming processing.

### 4.2 Topic analysis

By preprocessing the three fundamental datasets through R programming, we can get all the stemming words and their frequencies. The histograms of the most frequent words in emerging stage, slow growth stage, and rapid growth stage are shown in Figs 5–7.

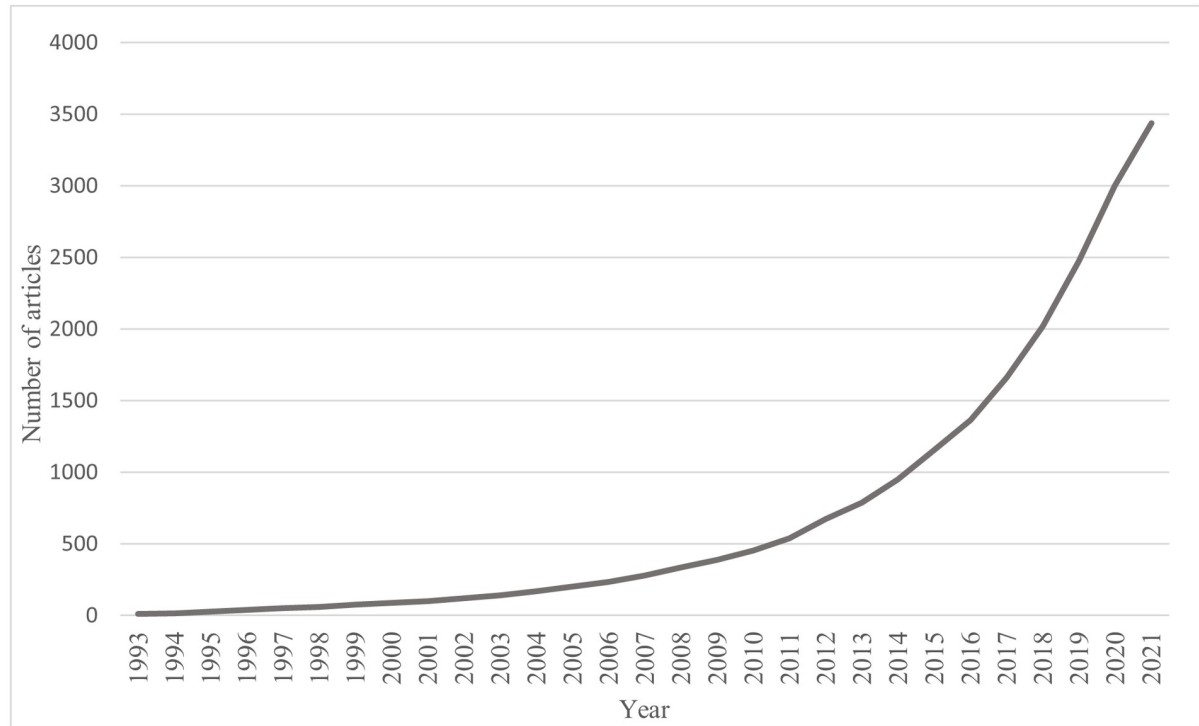

**Fig 4. Life cycle S-curve of green logistics.**

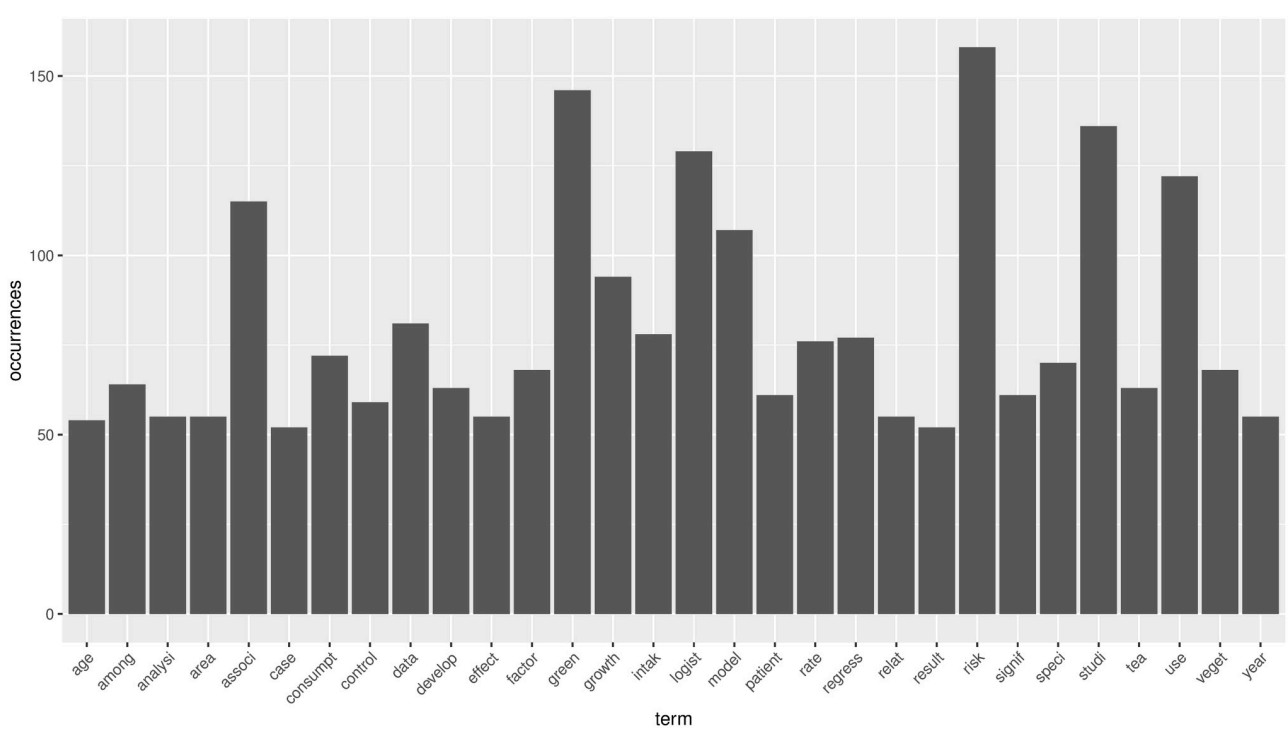

**Fig 5. The histogram of most frequency in emerging stage.**

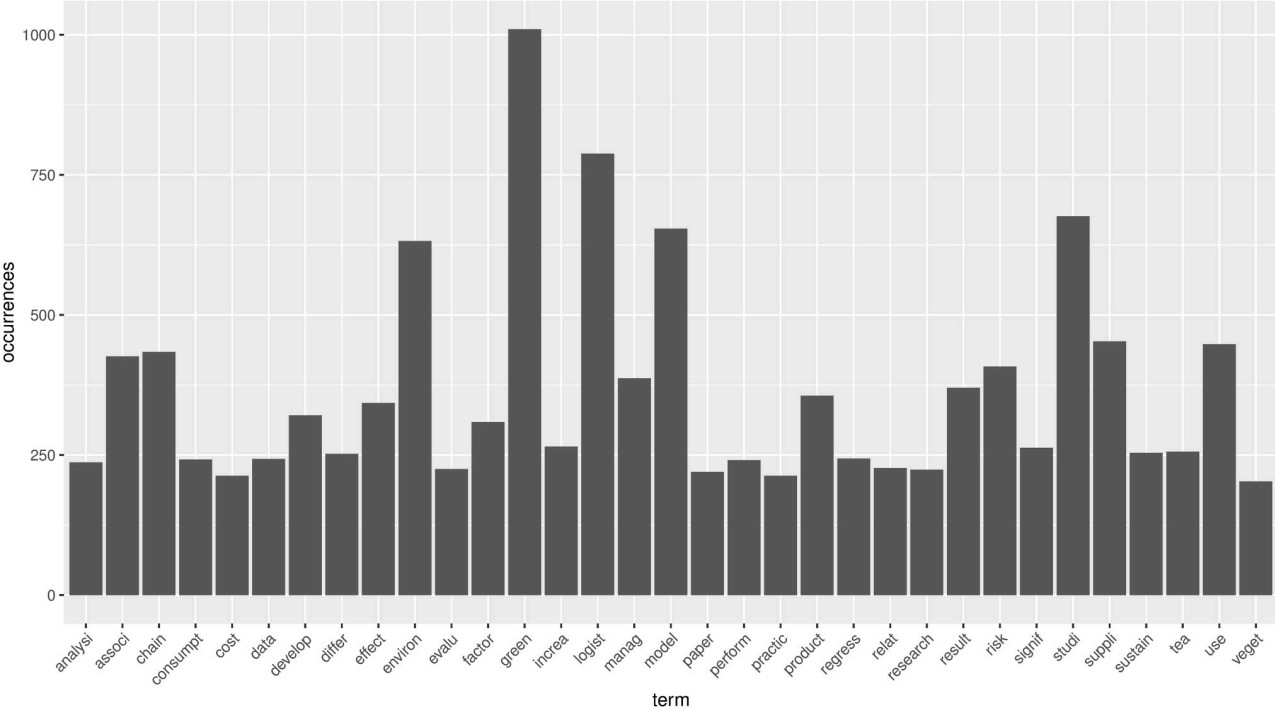

**Fig 6. The histogram of most frequency in slow growth stage.**

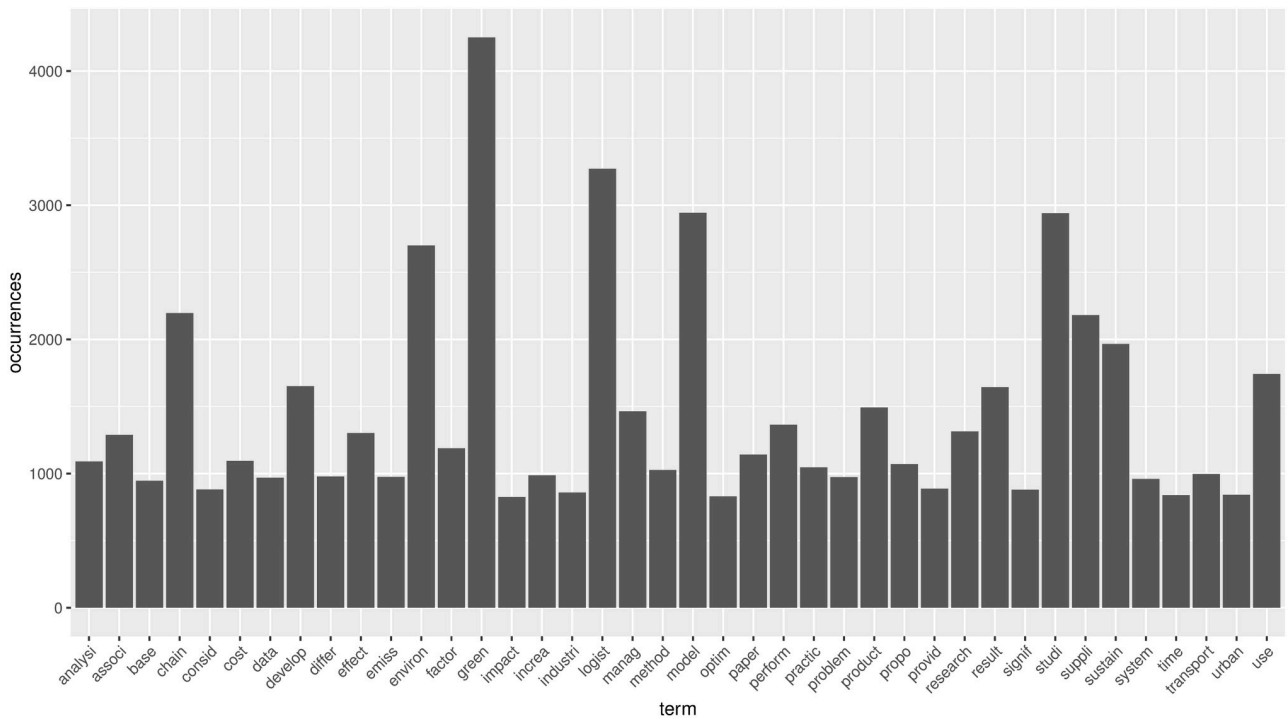

**Fig 7. The histogram of most frequency in rapid growth stage.**

**Table 2. Stem of words and label of the topics for emerging stage (1993–2003).**

| Topic number | Stem of words | Label of the topic |
|---|---|---|
| 1–1 | Risk; green; studi; logist; use; associ; model; growth; regress; intak; rate; speci; factor; data; develop | Elements of developing green logistics |
| 1–2 | Consumpt; environ; ratio; found; high; yield; portal; reduc; trend; treatment; adjust; describ; statist; measur; densiti | Trends in GL affected by over consumption and environmental improvements |
| 1–3 | Function; behavior; chang; examin; indic; role; fruit; mean; improv; obtain; presenc; simul; requir; develop; flower | The influencing factors of human behavior on GL |
| 1–4 | Relationship; purpo; produc; evalu; global; inform; product; anali; compar; less; data; leav; oper; base; effici | Relationship analysis between GL and other issues |

The most crucial parameter of the LDA topic model is the number of topics in each dataset. There are usually four types to determine the number of topics. First, we can get the number of topics by judgment based on subjective experience, and most people use this method. Second, the optimal number of topics under different time windows can be obtained by calculating the perplexity [63]. Third, some people determine the number of topics by the marginal likelihood function [64]. Fourth, there is a non-parametric method: the HDP method based on the Dirichlet process proposed by Teh (2005) [65]. In this paper, we choose the first method to calculate the topic numbers. We test the alternative topic numbers from 2 to 10 during the process and determine the optimal number of topics in each period. The optimal number of topics in the emerging stage is 4, the optimal number of topics in the slow growth stage is 5, and the optimal number of topics in the rapid growth stage is 7. We use 15 related stemming words to describe the meaning of each topic in each phase. The results are shown in Tables 2–4.

After collecting related stemming words of each topic in different stages, we can use the LDAvis package in R programming to visualize the results of LDA topic model. The results can be seen in Figs 8–10. The visualized pictures represent that every topic in each phase is a circle, where the number of circles is also the number of topics. If the loop is bigger, the quantity of text data is more extensive, and the topics are arranged by the amount. The lower left corner of the figure is an example of marginal topic distribution. The distance between various topics can indicate the degree of discrimination between them. These figures can describe the overall status of research and development activities of green logistics. Specifically, Fig 8 shows

**Table 3. Stem of words and label of the topics for slow growth stage (2004–2014).**

| Topic number | Stem of words | Label of the topic |
|---|---|---|
| 2–1 | Green; logist; environ; suppli; chain; model; manag; product; perform; practic; paper; research; sustain; cost; transport | Research on sustainable development management of GL |
| 2–2 | Associ; intak; speci; problem; impact; risk; Ltd; built; compani; econom; inver; express; approach; increa; among | Issues resulted by expanded logistic business |
| 2–3 | Studi; use; risk; associ; effect; model; factor; result; signif; vehicl; increa; regress; data; evalu; differ | Significant factors affecting logistics efficiency |
| 2–4 | Manufactur; consid; market; approach; deci; carbon; framework; literatur; vehicl; firm; analysi; busi; issu; general; method | The status quo of enterprise manufacturing and transportation |
| 2–5 | Closedloop; reserv; case; contain; gscm; life; port; solv Deci; reveal; valid; barrier; shipper; encrypt; issu | Closed loop SCM and green supply chain |

**Table 4. Stem of words and label of the topics for rapid growth stage (2015–2021).**

| Topic number | Stem of words | Label of the topic |
|---|---|---|
| 3–1 | Model; environ; green; studi; chain; suppli; logist; sustain; develop; manag; product; result; use; research; perform | Model and management on sustainable development of Green Logistics |
| 3–2 | Associ; studi; risk; analysi; factor; use; regress; data; model; urban; differ; among; signif; predict; chang | Related issues and potential threats of GL |
| 3–3 | Effect; optim; evalu; impact; factor; vehicl; improv; carbon; increa; process; approach; oper; differ; time; found | Optimization analysis of low-carbon vehicle routing and time |
| 3–4 | Logist; green; practic; perform; enterpri; rever; servic; oper; econom; rout; develop; distribut; gscm; compani; packag | The performance of logistic company in GSCM |
| 3–5 | Factor; valu; imag; intersect; adopt; import; yield; gas; reduct; network; local; quantit; sscm; perceiv; integr | Green supply chain network analysis |
| 3–6 | Gscm; recycl; signif; product; driver; ban; gsc; plant; pretreat; indic; strategi; effici; select; polici; measur | Strategy and policy of GSCM |
| 3–7 | Green; gscm; space; object; associ; barrier; servic; mental; data; yield; network; solut; road; gsc; disrupt | The dilemma of green logistics development |

the breakdown of the emerging stage according to LDA topic model. The numbers of circles in Fig 8 refer to topic 1–1 (elements of developing green logistics), topic 1–2 (trends in GL affected by over consumption and environmental improvements), topic 1–3 (the influencing factors of human behavior on GL) and topic 1–4 (relationship analysis between GL and other issues). The degree of discrimination between topic 1–3 and topic 1–4 is smaller than topic 1–1 and topic 1–2. As shown in Fig 9, topic 2–1 (research on sustainable development management of GL) and topic 2–2 (issues resulted by expanded logistic business) become central

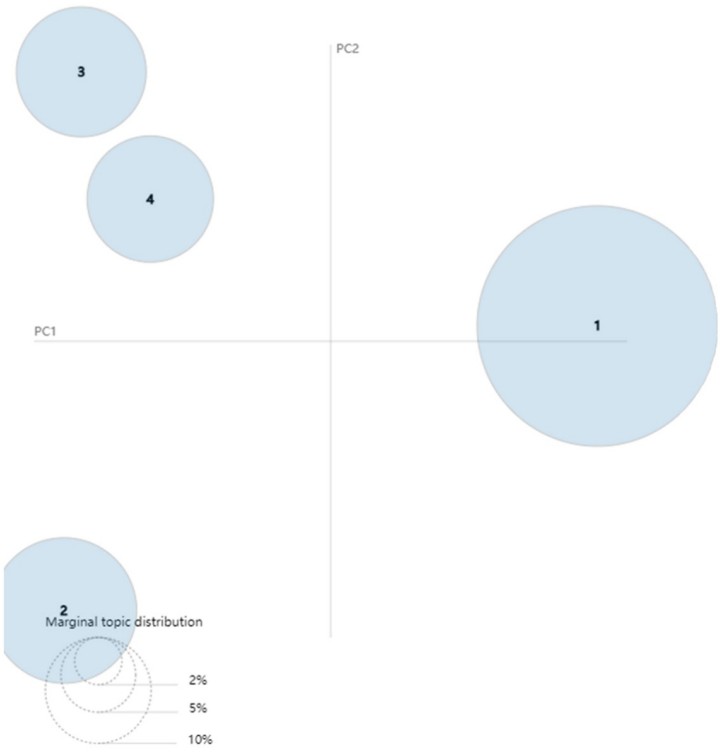

**Fig 8. Visualization of LDA topic model in emerging stage (1993–2003).**

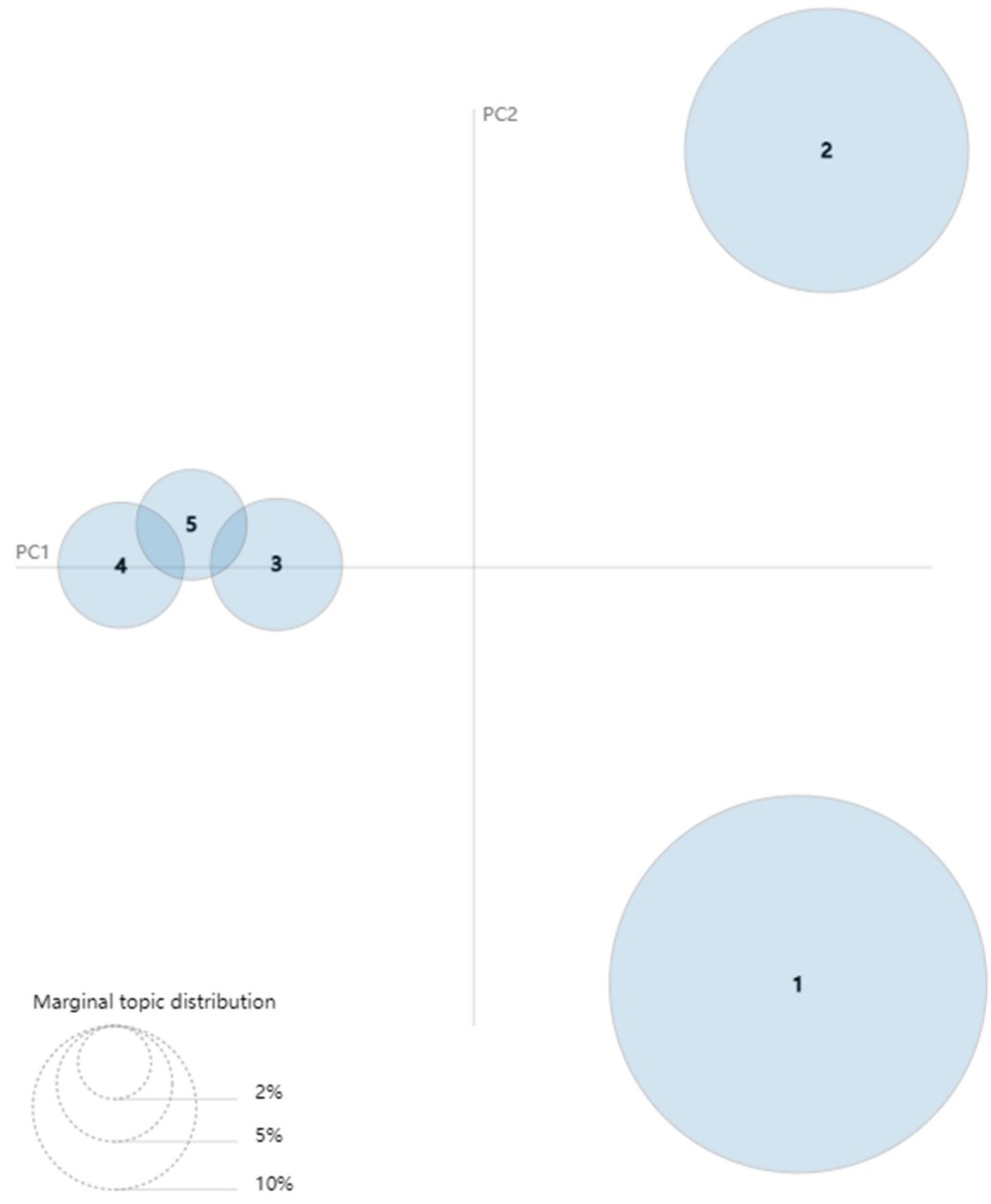

**Fig 9. Visualization of LDA topic model in slow growth stage (2004–2014).**

issues in slow growth stage. Both topic 2–3 (significant factors affecting logistics efficiency) and topic 2–4 (the status quo of enterprise manufacturing and transportation) share a few key features with topic 2–5 (Closed loop SCM and green supply chain). From Fig 10 we can see that there is a growing body of literature that recognizes topic 3–1 (model and management on sustainable development of GL, topic 3–2 (related issues and potential threats of GL) and topic 3–3 (optimization analysis of low-carbon vehicle routing and time) as mainstreams of green logistics in rapid growth stage. There are some similarities between topic 3–5 (green supply chain network analysis) and topic 3–6 (strategy and policy of GSCM).

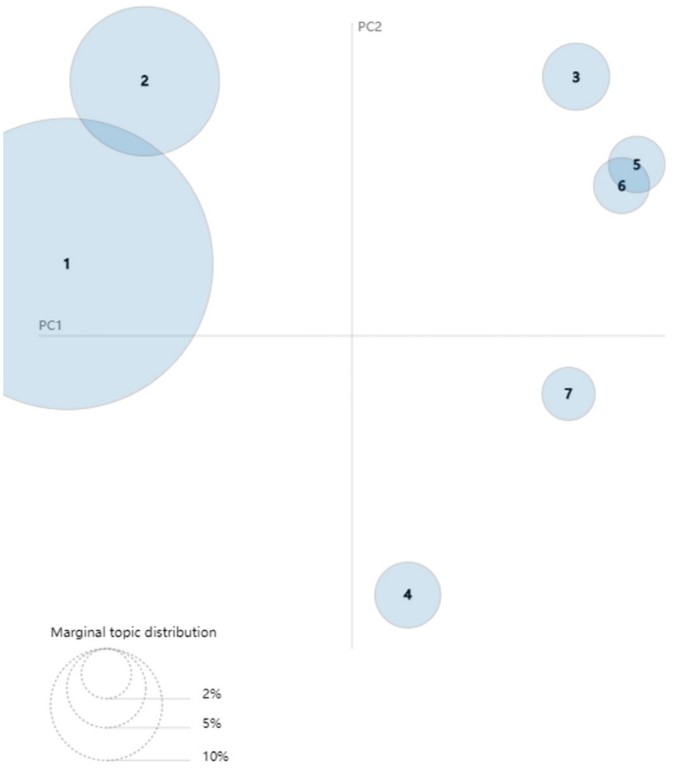

**Fig 10. Visualization of LDA topic model in rapid growth stage (2015–2021).**

## 4.3 Topic similarity calculation result and visualization

In this study, cosine similarity is used to calculate relationships between various hot topics, and then the evolutionary relationships between them are identified. The topic similarity matrix can be seen in Tables 5 and 6. We set a threshold value of 0.4 to determine the similarity between two topics. If the similarity is greater or equal to 0.4, there is a link between two topics. Based on the results of all calculation, we can get the final roadmap as like Fig 11.

Fig 11 describes the development roadmap of how green logistics evolved in past years. It shows the relevance between different topics in three stages, indicating the evolutionary mechanism of green logistics. As can be seen from Fig 11, the results show that the top three topics, including topic 3–1 (model and management on sustainable development of green logistics), topic 3–2 (related issues and potential threats of green logistics) and topic 3–3 (optimization analysis of low-carbon vehicle routing and time) are the hottest topics in recent years. The seven topics in rapid growth stage all evolved from emerging and slow growth stages. The specific process of topic evolution in green logistics is as follows.

**Table 5. Similarity between emerging stage and slow growth stage.**

| Topic | 2–1 | 2–2 | 2–3 | 2–4 | 2–5 |
|---|---|---|---|---|---|
| 1–1 | 0.63 | 0.54 | 0.455714 | 0.148571 | 0.118571 |
| 1–2 | 0.165714 | 0.438571 | 0.281429 | 0.124286 | 0.097143 |
| 1–3 | 0.186610 | 0.118234 | 0.267807 | 0.484046 | 0.171225 |
| 1–4 | 0.165478 | 0.122682 | 0.249644 | 0.495578 | 0.568474 |

**Table 6. Similarity between slow growth and rapid growth stage.**

| Topic | 3–1 | 3–2 | 3–3 | 3–4 | 3–5 | 3–6 | 3–7 |
|---|---|---|---|---|---|---|---|
| 2–1 | 0.652328 | 0.474178 | 0.136427 | 0.270202 | 0.409274 | 0.435765 | 0.142387 |
| 2–2 | 0.405965 | 0.308614 | 0.123182 | 0.090730 | 0.112585 | 0.107287 | 0.415897 |
| 2–3 | 0.518554 | 0.337096 | 0.495369 | 0.163580 | 0.141725 | 0.145036 | 0.147023 |
| 2–4 | 0.166229 | 0.424772 | 0.100002 | 0.488081 | 0.096029 | 0.478147 | 0.092717 |
| 2–5 | 0.251661 | 0.186759 | 0.400002 | 0.413910 | 0.201327 | 0.105300 | 0.115897 |

Topic 3–1 (model and management on sustainable development of green logistics) evolved from topic 2–1 (research on Sustainable Development Management of GL), topic 2–2 (issues resulted by expanded logistic business) and topic 2–3 (significant factors affecting logistics efficiency) in stage 2, topic 1–1 (elements of developing green logistics) and topic 1–2 (trends in GL affected by over consumption and environmental improvements) in stage 1. Topic 3–2 (related issues and potential threats of GL) evolved from topic 2–1 (research on Sustainable Development Management of GL) and topic 2–4 (the status quo of enterprise manufacturing and transportation) in stage 2, topic 1–1 (elements of developing GL), topic 1–3 (the Influencing factors of human behavior on GL) and topic 1–4 (relationship analysis between GL and other issues) in stage 1. Topic 3–3 (optimization analysis of low-carbon vehicle routing and time) evolved from topic 2–3 (significant factors affecting logistics efficiency) and topic 2–5 (closed loop SCM and green supply chain) in stage 2, topic 1–1 (elements of developing GL) and topic 1–4 (relationship analysis between GL and other issues) in stage 1. The above three topics in rapid growth stage are evolved from emerging and slow growth stages with a high degree of discussion. Moreover, the related topics in emerging stage and slow growth stage are also widely discussed and researched by scholars. Therefore, topic 3–1 (model and management on sustainable development of green logistics), topic 3–2 (related issues and potential threats of green logistics) and topic 3–3 (optimization analysis of low-carbon vehicle routing and time) are the critical topics in green logistics field.

Topic 3–4 (the performance of logistic company in GSCM) evolved from topic 2–4 (the status quo of enterprise manufacturing and transportation) and topic 2–5 (closed loop SCM and green supply chain) in stage 2, and topic 1–3 (the influencing factors of human behavior on

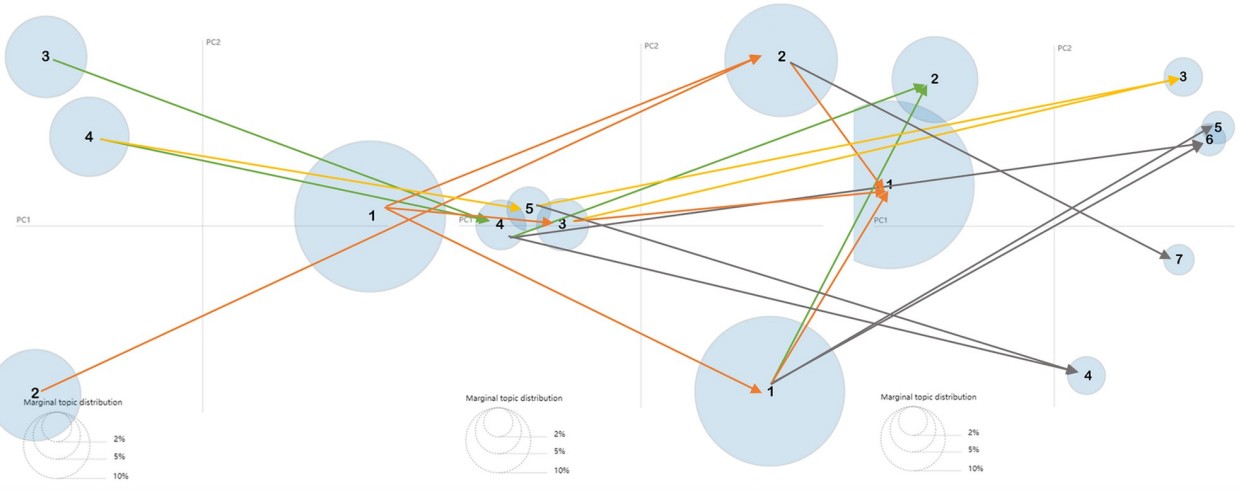

**Fig 11. LDA-based roadmap of green logistics.**

GL) and topic 1–4 (relationship analysis between GL and other issues) in stage 1. Topic 3–5 (green supply chain network analysis) evolved from topic 2–1 (research on sustainable development management of GL) in stage 2 and topic 1–1 (elements of developing green logistics) in stage 1. The evolution of above two topics presents that the scholars try to conduct a more objective and practical analysis of green logistics. It can effectively promote the modernization of green logistics.

Topic 3–6 (strategy and policy of GSCM) evolved from topic 2–1 (research on sustainable development management of GL) and topic 2–4 (the status quo of enterprise manufacturing and transportation) in stage 2, topic 1–1 (elements of developing GL), topic 1–3 (the influencing factors of human behavior on GL) and topic 1–4 (relationship analysis between GL and other issues) in stage 1. Topic 3–7 (the dilemma of green logistics development) evolved from topic 2–2 (issues resulted by expanded logistic business) in stage 2, topic 1–1 (elements of developing green logistics) and topic 1–2 (trends in GL affected by over consumption and environmental improvements) in stage 1. The degree of discussion in topic 3–6 (strategy and policy of GSCM) and topic 3–7 (the dilemma of green logistics development) seems to have declined in the current stage. It doesn't mean that the laws and regulations of green logistics and its predicament are unnecessary. By contrast, people need to pay more attention to the current dilemma of green logistics and constantly improve the strategy and policy related to green logistics.

## 5. Conclusions and limitations

This paper analyzes the current research and development status and evolution of green logistics based on LDA topic model. We divide the development of green logistics into three stages, which are emerging stage (1993–2003), slow growth stage (2004–2014), and rapid growth stage (2015–2021). Based on LDA topic model and text similarity, we get the four topics in emerging stage, five in slow growth stage and seven in rapid growth stage. Furthermore, the relevance was calculated between different topics in different phases by cosine similarity. Finally, the roadmap showing the evolution process of green logistics can be obtained. Seven topics in rapid growth stage are evolved from five topics in slow growth stage and four topics in emerging stage. The findings of this study suggest that the mainstreams of the current development of green logistics include "model and management on sustainable development of green logistics", "related issues and potential threats of green logistics", and "optimization analysis of low-carbon vehicle routing and time". "The performance of logistic company in GSCM", "green supply chain network analysis", "strategy and policy of GSCM" and "the dilemma of green logistics development" are important points that people will continue to study and pay attention to in the future. The evolution analysis of green logistics proves that scholars have conducted more comprehensive and specific research on the development of green logistics in recent years. Taken together, while continuously strengthening the technology of green logistics, scholars also need to propose corresponding effective strategies and objectively analyze the current dilemma of green logistics to optimize the entire green logistics process.

There are several limitations that can be addressed in future studies. This study collects the article data only from Web of Science. The cosine similarity method is not perfect for analyzing the semantic similarity of different topics, and some parameters in the LDA topic model need to be recognized by authoritative experts. Future studies can consider collecting more articles from other datasets like Scopus and ScienceDirect, and exploring a new method to replace the cosine similarity method. The code of LDA topic model by R programming can also be improved in the future.

Our study contributes a new perspective on how to analyze the evolution of green logistics and lays the groundwork for future research into the main directions of green logistics. This type of analysis can be of value for the industry development roadmapping and foresight, and our method offers improvements to the current related studies.

## Author Contributions

**Conceptualization:** Yong Jin Kim.

**Formal analysis:** Ruijundi Ma.

**Methodology:** Ruijundi Ma.

**Project administration:** Yong Jin Kim.

**Supervision:** Yong Jin Kim.

**Writing – original draft:** Ruijundi Ma.

**Writing – review & editing:** Yong Jin Kim.

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
