## [Decision Letter · Decision Letter 0]

29 May 2023

PONE-D-23-13824Tracing the evolution of green logistics: A latent dirichlet allocation based topic modeling technology and roadmappingPLOS ONE

Dear Dr. Kim,

Thank you for submitting your manuscript to PLOS ONE. After careful consideration, we feel that it has merit but does not fully meet PLOS ONE’s publication criteria as it currently stands. Therefore, we invite you to submit a revised version of the manuscript that addresses the points raised during the review process.

We look forward to receiving your revised manuscript.

Kind regards,

Ibrahim Badi, PhD

Academic Editor

PLOS ONE

Journal Requirements:

"This work is supported by the Korea Agency for Infrastructure Technology Advancement (KAIA) grant funded by the Ministry of Land, Infrastructure and Transport (Grant 21HCLP-C161726-01)".

"NO authors have competing interests".

6. We note that Figures 6,8 and 10 in your submission contain copyrighted images. All PLOS content is published under the Creative Commons Attribution License (CC BY 4.0), which means that the manuscript, images, and Supporting Information files will be freely available online, and any third party is permitted to access, download, copy, distribute, and use these materials in any way, even commercially, with proper attribution. For more information, see our copyright guidelines: http://journals.plos.org/plosone/s/licenses-and-copyright.

a. You may seek permission from the original copyright holder of Figures 6,8 and 10 to publish the content specifically under the CC BY 4.0 license. 

Reviewers' comments:

Reviewer's Responses to Questions

**Comments to the Author**

1. Is the manuscript technically sound, and do the data support the conclusions?

Reviewer #1: Yes

Reviewer #2: Yes

2. Has the statistical analysis been performed appropriately and rigorously? 

Reviewer #1: Yes

Reviewer #2: Yes

3. Have the authors made all data underlying the findings in their manuscript fully available?

Reviewer #1: No

Reviewer #2: No

4. Is the manuscript presented in an intelligible fashion and written in standard English?

Reviewer #1: Yes

Reviewer #2: No

5. Review Comments to the Author

Reviewer #1: This paper analyzed and reviewed green logistics topics and their evolution. Green Logistics (GL) is the trend of modern logistics, the topic is meaningful, and the manuscript in its current form provides a good balance of necessary background information, methodologies, and results. However, I feel that the article can benefit from many amendments as described below.

1. The figures in the manuscript need improvement.

2. The structure of the literature review section in the manuscript is not appropriate and more literature should be added. The following references are suggested for consideration:

[1] The impact of carbon emission trading policy on firms’ green innovation in China；[2] Evaluating R&D efficiency of China’s listed lithium battery enterprises; [3] Government regulation to promote coordinated emission reduction among enterprises in the green supply chain based on evolutionary game analysis；[4] Low carbon technology for carbon neutrality in sustainable cities: A survey.

3. The contribution of this manuscript is not clearly described and lacks organization.

4. Limited comparison with existing algorithms, it is recommended to use other models to compare and contrast with the LDA model.

Reviewer #2: Thank you for the opportunity to read and review the paper titled Tracing the evolution of green logistics: A latent dirichlet allocation based topic modeling technology and roadmapping. The topic of the paper is important for international debate. However, the article suffers from several limitations that makes it not acceptable in a present form.

Abstract

In the abstract, it should briefly mention a research gap.

Introduction

The beginning of the text (lines 34-57) is somewhat chaotic and inconsistent. Many threads regarding specific areas of green logistics activities emerge. Only at the end of this passage do we learn that the authors intend to conduct an analysis of green logistics from its macroeconomic point of view.

I suggest reorganizing the text to be more logical.

I think it is better to develop the gap more deeply in the introduction along with the objectives, the mention of the contribution and the organization of the work in the Introduction section.

Literature review section

In this section, there is a similar problem with the logical reasoning. A reader who is not familiar with the topic of green logistics will have difficulty understanding the text. The authors refer to the main components of green logistics, but they do not explain earlier why. Each sentence should result from the previous one.

Roadmapping analysis

The text does not provide sufficient justification for why analyzing the evolution of green logistics and visualizing its development is a new perspective or how it differs from existing approaches.

Methodology:

The text lacks a clear explanation of why the Web of Science database was chosen as the data source and how it ensures comprehensive coverage of green logistics development.

The text does not provide sufficient detail on how the division of green logistics development phases is conducted based on the life cycle theory.

General comments

At the end of the article, the diagrams are not numbered, which significantly hinders the analysis of the text.

Style of English: The Latent Dirichlet Allocation (LDA) model, as the current mainstream topic model, is widely used in topic mining to mine latent topics in the initial document. It can be used another word than repeat it twice.

The sentence (lines 422-434): This article divides the development of green logistics into three stages, which are emerging stage (1993-2003), slow growth stage (2004-2014), and rapid growth stage (2015-2021). I suggest rephrasing this sentence, as it is not the article itself that divides logistics into three phases, but rather the authors, or it is divided based on specific criteria.

6. PLOS authors have the option to publish the peer review history of their article (what does this mean?). If published, this will include your full peer review and any attached files.

Reviewer #1: No

Reviewer #2: No

---

## [Author Response · Author response to Decision Letter 0]

24 Jul 2023

Dear Editor and Reviewers,

Thank you for giving me the opportunity to submit a revised draft of my manuscript titled Tracing the evolution of green logistics: A latent dirichlet allocation based topic modeling technology and roadmapping to PLOS ONE. We appreciate the time and effort that you have dedicated to providing your valuable feedback on our manuscript. We are grateful to the reviewers for your insightful comments on our paper. We have been able to incorporate changes to reflect most of the suggestions provided by the reviewers. We have highlighted the changes within the manuscript.

Here is a point-by-point response to the editor and reviewers’ comments and concerns.

Comments from Reviewer 1

-Comment 1: The figures in the manuscript need improvement.

Response: Thank you for pointing this out. We agree with this and have improved all figures in the manuscript, especially the clarity.

-Comment 2: The structure of the literature review section in the manuscript is not appropriate and more literature should be added. The following references are suggested for consideration:

[1] The impact of carbon emission trading policy on firms’ green innovation in China；[2] Evaluating R&D efficiency of China’s listed lithium battery enterprises; [3] Government regulation to promote coordinated emission reduction among enterprises in the green supply chain based on evolutionary game analysis；[4] Low carbon technology for carbon neutrality in sustainable cities: A survey.

Response: We are grateful for the suggestion. As suggested by reviewer, we have restructured our literature review, mainly reflected in the “status quo of green logistics” and “roadmapping analysis” parts. We provided more details to describe why we should analyze the evolution of green logistics and how we address our gap in this research. In addition, more papers were discussed in the literature review, including the articles mentioned by the reviewer, which can be found in lines 130-140. These articles complement this study from the perspective of low-carbon environmental protection and policy support.

-Comment 3: The contribution of this manuscript is not clearly described and lacks organization.

Response: Thank you for the advice. We have supplemented more details to describe the contribution of this paper in lines 487-491. In addition, the contribution of this study also can be found in the abstract (lines 27-30) and introduction part (lines 84-86).

-Comment 4: Limited comparison with existing algorithms, it is recommended to use other models to compare and contrast with the LDA model.

Response: Thank you for your nice comment. According to your suggestion, we have included additional information discussing the comparison between the LDA model and other models. We emphasized the advantages of the LDA method in green logistics evolution analysis from two dimensions after comparing it with other methods. The revised parts can be seen in lines 245-256. Moreover, we also complemented the literature review part about the LDA topic model, which can be found in lines 147-152.

Comments from Reviewer 2

-Comment 1: In the abstract, it should briefly mention a research gap.

Response: Thank you for pointing this out. We agree with this comment. We have emphasized the research gap briefly in the abstract, as seen in the manuscript lines 27-30.

-Comment 2: The beginning of the text (lines 34-57) is somewhat chaotic and inconsistent. Many threads regarding specific areas of green logistics activities emerge. Only at the end of this passage do we learn that the authors intend to conduct an analysis of green logistics from its macroeconomic point of view.

I suggest reorganizing the text to be more logical.

I think it is better to develop the gap more deeply in the introduction along with the objectives, the mention of the contribution and the organization of the work in the Introduction section.

Response: Thank you for your comment. We agree. Therefore, we have reorganized the introduction, especially the beginning of the text (lines 34-57). We develop the gap deeply in the research on the evolution of green logistics, which can be found in the second paragraph (lines 34-56). Furthermore, the contribution of this research is emphasized in both introduction (lines 73-86) and conclusion parts (lines 487-491).

-Comment 3: Literature review section

In this section, there is a similar problem with the logical reasoning. A reader who is not familiar with the topic of green logistics will have difficulty understanding the text. The authors refer to the main components of green logistics, but they do not explain earlier why. Each sentence should result from the previous one.

Response: We sincerely appreciate the reviewer’s suggestion. According to the reviewer’s comments, we have reorganized the structure of the literature review part related to green logistics to make our expression more precise and comprehensive, which can be seen in lines 92-140. We have added a more detailed interpretation regarding how green logistics developed and the main components of green logistics.

-Comment 4: Roadmapping analysis

The text does not provide sufficient justification for why analyzing the evolution of green logistics and visualizing its development is a new perspective or how it differs from existing approaches.

Response: Thank you for your comment. More details on why analyzing the evolution of green logistics have been argued in “status quo of green logistics” part (lines 130-140). And we have reorganized “roadmapping analysis” part to explain why we use roadmapping to analyze the evolution of green logistics (lines 170-188). The roadmapping approach is a mature method to visualize the process of a technology or a research objective. Thus, we introduce this method to analyze the evolution of green logistics directly rather than compare it with other approaches. Furthermore, the new perspective proposed by this study is that this paper tries to introduce LDA topic model and roadmapping to analyze the evolution of green logistics based on article data.

-Comment 5: Methodology

The text lacks a clear explanation of why the Web of Science database was chosen as the data source and how it ensures comprehensive coverage of green logistics development.

Response: Agree. We have, accordingly, modified the contents of the first paragraph in the methodology part to emphasize this point. We describe the advantages and coverage of Web of Science (WoS) and explain why we used WoS as a database to collect original article data. The revised part can be seen on lines 198-204.

-Comment 6: The text does not provide sufficient detail on how the division of green logistics development phases is conducted based on the life cycle theory.

Response: Thank you for the suggestion. We have supplemented more details on how the green logistics development phase is divided based on the life cycle theory and explained why we use life cycle theory and S curve to analyze the evolution of green logistics. The modified part can be found in the methodology part (lines 228-233) and the experiment and results part (lines 314-322). 

General comments

-Comment 1: At the end of the article, the diagrams are not numbered, which significantly hinders the analysis of the text.

Response: I can not clearly understand this comment. I just improve the readability by improving the quality of figures. If you give me an additional comment I will edit the manuscript.

-Comment 2: Style of English: The Latent Dirichlet Allocation (LDA) model, as the current mainstream topic model, is widely used in topic mining to mine latent topics in the initial document. It can be used another word than repeat it twice.

Response: Thank you for the comment. We apologize for the language problems in the original manuscript. The similar language expression has been improved.

-Comment 3: The sentence (lines 422-434): This article divides the development of green logistics into three stages, which are emerging stage (1993-2003), slow growth stage (2004-2014), and rapid growth stage (2015-2021). I suggest rephrasing this sentence, as it is not the article itself that divides logistics into three phases, but rather the authors, or it is divided based on specific criteria.

Response: We accept this comment. The sentence (lines 422-434) mentioned by the reviewer has been rephrased, which can be seen in lines 459-474.

In addition to the above comments, Figure 6, 8 and 10 mentioned by Journal Requirements have been removed.

We look forward to hearing from you in due time regarding our submission and to respond to any further questions and comments you may have. 

Sincerely,

Yong Jin Kim

---

## [Decision Letter · Decision Letter 1]

2 Aug 2023

Tracing the evolution of green logistics: A latent dirichlet allocation based topic modeling technology and roadmapping

PONE-D-23-13824R1

Dear Dr. Kim,

We’re pleased to inform you that your manuscript has been judged scientifically suitable for publication and will be formally accepted for publication once it meets all outstanding technical requirements.

Kind regards,

Ibrahim Badi, PhD

Academic Editor

PLOS ONE

Additional Editor Comments (optional):

Reviewers' comments:

Reviewer's Responses to Questions

**Comments to the Author**

1. If the authors have adequately addressed your comments raised in a previous round of review and you feel that this manuscript is now acceptable for publication, you may indicate that here to bypass the “Comments to the Author” section, enter your conflict of interest statement in the “Confidential to Editor” section, and submit your "Accept" recommendation.

Reviewer #1: All comments have been addressed

2. Is the manuscript technically sound, and do the data support the conclusions?

Reviewer #1: Yes

3. Has the statistical analysis been performed appropriately and rigorously? 

Reviewer #1: Yes

4. Have the authors made all data underlying the findings in their manuscript fully available?

Reviewer #1: Yes

5. Is the manuscript presented in an intelligible fashion and written in standard English?

Reviewer #1: Yes

6. Review Comments to the Author

Reviewer #1: The authors have addressed all my concerns, the quality has been improved, and the current version can be accepted.

7. PLOS authors have the option to publish the peer review history of their article (what does this mean?). If published, this will include your full peer review and any attached files.

Reviewer #1: No

---

## [Editor Report · Acceptance letter]

7 Aug 2023

PONE-D-23-13824R1 

Tracing the evolution of green logistics: A latent dirichlet allocation based topic modeling technology and roadmapping 

Dear Dr. Kim:

I'm pleased to inform you that your manuscript has been deemed suitable for publication in PLOS ONE. Congratulations! Your manuscript is now with our production department. 

Kind regards, 

on behalf of

Dr. Ibrahim Badi 

Academic Editor

PLOS ONE